# A range three elliptic deformation of the Hubbard model

Marius de Leeuw[1], Chiara Paletta[1], Balázs Pozsgay[2]

January 13, 2023

[1]School of Mathematics & Hamilton Mathematics Institute, Trinity College Dublin, Ireland
[2]MTA-ELTE "Momentum" Integrable Quantum Dynamics Research Group, Department of Theoretical Physics, Eötvös Loránd University, Budapest, Hungary

mdeleeuw@maths.tcd.ie, palettac@maths.tcd.ie, pozsgay.balazs@gmail.com

**Abstract**

In this paper we present a new integrable deformation of the Hubbard model. Our deformation gives rise to a range 3 interaction term in the Hamiltonian which does not preserve spin or particle number. This is the first non-trivial medium range deformation of the Hubbard model that is integrable. Our model can be mapped to a new integrable nearest-neighbour model via a duality transformation. The resulting nearest-neighbour model also breaks spin conservation. We compute the $R$-matrices for our models, and find that there is a very unusual dependence on the spectral parameters in terms of the elliptic amplitude.

## 1 Introduction

The Hubbard model describes the physics of interacting spin-1/2 fermions on the lattice, and it is one of the most important models in the condensed matter literature. In one space dimension it is exactly solvable by the Bethe Ansatz [1, 2], enabling the exact computation of interesting phenomena such as spin-charge separation. The model is integrable and it can be embedded into the standard framework of the Yang-Baxter equation; this is achieved using the $R$-matrix of Shastry [3]. The transport properties of the model have been an object of interest for many decades (see for example [4]), and research in this direction is still ongoing [5, 6, 7, 8, 9]. Recently so-called integrable quantum quenches have also been considered in the 1D Hubbard model [10], using information also about exact overlaps [11, 12]. The recent work [13] derived explicit expression for all local charges of the model.

The Hubbard model is also important for research on the AdS/CFT correspondence [14]. It turns out that the $R$-matrix, which is relevant for the AdS/CFT correspondence, is related to Shastry's $R$-matrix [15, 16]. This remarkable relation shed some new light on the symmetry algebra of the Hubbard model. It was known for a long time that the Hubbard model exhibits $SU(2) \times SU(2)$ symmetry [17, 18]. By using the map to string theory these could be seen as coming from a centrally extended superalgebra from which the Hubbard model can be obtained in a certain limit [19]. Moreover, this observation recently lead to the formulation of the so-called quantum spectral curve for the Hubbard model [20].

Over the years, many extensions and generalizations of the Hubbard model appeared, and many of the models were found to be integrable. Examples include the models found from the $R$-matrix of Shastry [21, 22], the models of Bariev and Alcaraz [23] (see also [24]), the Essler-Korepin-Schoutens model, [25], and multi-component generalizations [26, 27, 28].

In this paper we consider a new extension of the Hubbard model. Our model belongs to the class of medium range spin chains: it has next-to-nearest-neighbour interactions and it is still integrable. The model depends on two parameters: the Hubbard interaction strength and a deformation parameter. If both parameters are real, the model is Hermitian. The deformation violates both the spin and charge conservation, therefore our model is reminiscent of the XYZ spin chain. Accordingly, we find that the $R$-matrix is elliptic. However the dependence of the $R$-matrix on the spectral parameter in very unusual.

We furthermore find that the new model can be transformed into a spin chain with nearest-neighbour interactions after applying a certain duality (or bond-site) transformation. This model is also characterized by two parameters, whose reality determines the Hermiticity of the model. However, after the transformation there is no direct connection to the Hubbard model.

The paper is structured as follows. In Section 2, we will first briefly discuss the Hubbard model, with both the fermionic and the bosonic formulations, and the symmetries. After this, in Section 3 we introduce the 3-site extension of the Hubbard model and show that it is integrable. In Section 4, we introduce the bond-site transformation and show that our model becomes a new integrable model with nearest-neighbour interactions. In Section 5 we prove the integrability properties of our model; the explicit form of the $R$-matrix is presented in the Appendix A. Finally, in Section 6 we discuss the large coupling limit of the models.

## 2  The Hubbard model

In this section, we give the basic definition of the Hubbard model. We also discuss several transformations and reformulations to bring it into a form, which is more convenient for our later purposes. We also briefly discuss the symmetries of the Hubbard model.

**Definition**   Let us consider a fermionic Hilbert space, with two species of particles which can be identified with electrons with spin up and down. We use the standard fermionic creation and annihilation operators $(c_j^{\uparrow,\downarrow})^\dagger$, $c_j^{\uparrow,\downarrow}$, which satisfy the canonical anti-commutation relations

$$\begin{aligned}
\{c_j^\alpha, c_k^\beta\} &= 0, \qquad \alpha, \beta = \uparrow, \downarrow \\
\{c_j^\alpha, (c_k^\beta)^\dagger\} &= \delta^{\alpha,\beta}\delta_{j,k},
\end{aligned} \tag{2.1}$$

where $j, k$ refer to the local Hilbert spaces.

We will also use the local particle number operators $n_j^\alpha = c_j^{\alpha\dagger}c_j^\alpha$. The local Hilbert space is spanned by the four vectors

$$|\emptyset\rangle, \qquad |\uparrow\rangle = (c^\uparrow)^\dagger|\emptyset\rangle, \qquad |\downarrow\rangle = (c^\downarrow)^\dagger|\emptyset\rangle, \qquad |\updownarrow\rangle = (c^\downarrow)^\dagger(c^\uparrow)^\dagger|\emptyset\rangle. \tag{2.2}$$

The Hubbard model [2, 29] is defined by the Hamiltonian

$$H = \sum_j \left[ (c_j^\uparrow)^\dagger c_{j+1}^\uparrow + (c_{j+1}^\uparrow)^\dagger c_j^\uparrow + (c_j^\downarrow)^\dagger c_{j+1}^\downarrow + (c_{j+1}^\downarrow)^\dagger c_j^\downarrow + U n_j^\uparrow n_j^\downarrow \right], \tag{2.3}$$

where $U \in \mathbb{R}$ is the coupling constant of the model. We will consider the model with both periodic and free boundary conditions. In the periodic case it is understood that the sum over $j$ runs from 1 to $L$ with the identification $L + 1 \equiv 1$, whereas in the case of free boundary conditions $j$ runs from 1 to $L - 1$.

The model has particle number conservation for both species separately. Hence the Hamiltonian commutes with the "total particle number" $N$ and the "total spin" $S_z$ defined as

$$N = \sum_j n_j^\uparrow + n_j^\downarrow, \qquad\qquad S_z = \sum_j n_j^\uparrow - n_j^\downarrow. \tag{2.4}$$

Therefore, it is possible to add two magnetic fields. A convenient choice is to add magnetic fields so that the interaction term becomes particle/hole symmetric. This choice preserves the integrability of the model and its explicit form is

$$H' = \sum_j \left[ (c_j^\uparrow)^\dagger c_{j+1}^\uparrow + (c_{j+1}^\uparrow)^\dagger c_j^\uparrow + (c_j^\downarrow)^\dagger c_{j+1}^\downarrow + (c_{j+1}^\downarrow)^\dagger c_j^\downarrow + \frac{U}{4}(1 - 2n_j^\uparrow)(1 - 2n_j^\downarrow) \right]. \tag{2.5}$$

This Hamiltonian enjoys $SU(2) \times SU(2)$ symmetry; the symmetry properties will be discussed in more detail below.

**Spin chain formulation**   For our purposes it is convenient to work with the "bosonic" version of the model. In order to do this, we perform an (inverse) Jordan-Wigner transformation to commuting spin chain operators. The operation can be performed in the case of open boundary conditions. The local Hilbert space is the tensor product

$$V_j = \mathbb{C}^2 \otimes \mathbb{C}^2 \tag{2.6}$$

with the full Hilbert space being the tensor product

$$V = \otimes_{j=1}^L V_j, \tag{2.7}$$

with $L$ the length of the spin chain. Using a standard notation in the literature, we introduce two sets of Pauli matrices $\sigma^a$ and $\tau^a$, $a = x, y, z$ that act respectively in the first or in the second copy of $\mathbb{C}^2$. The connection between the operators is

$$\sigma_j^- = \left[ \prod_{k=1}^{j-1} (-1)^{n_k^\uparrow} \right] c_j^\uparrow, \qquad\qquad \tau_j^- = \left[ \prod_{k=1}^{j-1} (-1)^{n_k^\downarrow} \right] c_j^\downarrow, \tag{2.8}$$

$$\sigma_j^+ = (c_j^\uparrow)^\dagger \left[ \prod_{k=1}^{j-1} (-1)^{n_k^\uparrow} \right], \qquad\qquad \tau_j^+ = (c_j^\downarrow)^\dagger \left[ \prod_{k=1}^{j-1} (-1)^{n_k^\downarrow} \right], \tag{2.9}$$

$$\sigma_j^z = 1 - 2n_j^\uparrow, \qquad\qquad \tau_j^z = 1 - 2n_j^\downarrow. \tag{2.10}$$

This transforms the Hubbard model Hamiltonian (2.5) to its bosonic formulation

$$H'' = \sum_j \left[ \sigma_j^+ \sigma_{j+1}^- + \sigma_j^- \sigma_{j+1}^+ + \tau_j^+ \tau_{j+1}^- + \tau_j^- \tau_{j+1}^+ + \frac{U}{4} \sigma_j^z \tau_j^z \right], \tag{2.11}$$

where $U$ is still the coupling constant of the model. At $U = 0$ the model describes two independent XX spin chains which do not interact with each other.

Let us now consider the model with periodic boundary conditions and volume $L = 4k$, $k \in \mathbb{N}$. In this case, we can perform a similarity transformation by the diagonal operator

$$D = D^\sigma D^\tau, \tag{2.12}$$

with

$$D^\sigma = \otimes_{j=1}^L \left[ \begin{pmatrix} i^j & 0 \\ 0 & 1 \end{pmatrix} \otimes 1_2 \right] = i^k \exp\left[ \sum_j \frac{i\pi j}{4} \sigma_j^z \right], \tag{2.13}$$

$$D^\tau = \otimes_{j=1}^L \left[ 1_2 \otimes \begin{pmatrix} i^j & 0 \\ 0 & 1 \end{pmatrix} \right] = i^k \exp\left[ \sum_j \frac{i\pi j}{4} \tau_j^z \right], \tag{2.14}$$

$1_2$ is the $2 \times 2$ Identity matrix.

Then we obtain

$$H_1 \equiv D^{-1} H'' D = \sum_j \left[ h^\sigma_{j,j+1} + h^\tau_{j,j+1} + \frac{U}{4} \sigma^z_j \tau^z_j \right], \tag{2.15}$$

where

$$h^\sigma_{j,j+1} \equiv i \left( \sigma^+_j \sigma^-_{j+1} - \sigma^-_j \sigma^+_{j+1} \right), \qquad h^\tau_{j,j+1} \equiv i \left( \tau^+_j \tau^-_{j+1} - \tau^-_j \tau^+_{j+1} \right). \tag{2.16}$$

The notation $H_1$ for the Hamiltonian signals that *the interaction term is a one-site operator*. Later we will also introduce Hamiltonians $H_k$ with $k = 2, 3$. Our convention will be the same: $H_k$ is a Hamiltonian where the kinetic term is a standard two-site hopping term, but the interaction term spans $k$ sites.

The kinetic terms above are known as "Dzyaloshinskii–Moriya interaction" terms [30], which becomes apparent after the rewriting

$$h^\sigma_{j,j+1} = \frac{1}{2} \left[ \sigma^x_j \sigma^y_{j+1} - \sigma^y_j \sigma^x_{j+1} \right], \tag{2.17}$$

and similarly for $h^\tau_{j,j+1}$. These hopping terms are antisymmetric with respect to space reflection.

If the volume $L$ is divisible by 4, then model Hamiltonians (2.11) and (2.15) are completely equivalent, despite the apparent spatial asymmetry. However, the Hamiltonian (2.15) defines an integrable model in itself, and we take this model as the starting point of our discussion.

**Symmetries** Now we discuss the symmetries of the Hubbard model in more detail, focusing on the Hamiltonian (2.15). The Hubbard model has both continuous as well as discrete symmetries.

For what follows we would like to introduce the so-called Shiba transformation [2]. It is defined on a chain of even length $L$ by

$$\begin{aligned} \mathcal{S}^\sigma &= \sigma^y_L \sigma^x_{L-1} \dots \sigma^y_2 \sigma^x_1, \\ \mathcal{S}^\tau &= \tau^y_L \tau^x_{L-1} \dots \tau^y_2 \tau^x_1. \end{aligned} \tag{2.18}$$

A similarity transformation with either $\mathcal{S}^\sigma$ or $\mathcal{S}^\tau$ preserves the kinetic term of $H_1$, while changing the sign of the interaction term. Explicitly,

$$S^\sigma H_1 S^\sigma = S^\tau H_1 S^\tau = \sum_j \left[ h^\sigma_{j,j+1} + h^\tau_{j,j+1} - \frac{U}{4} \sigma^z_j \tau^z_j \right]. \tag{2.19}$$

As a result, the combination of the two Shiba transformations is a discrete symmetry:

$$\mathcal{S}^\tau \mathcal{S}^\sigma H_1 \mathcal{S}^\sigma \mathcal{S}^\tau = H_1. \tag{2.20}$$

The Hubbard model Hamiltonian also enjoys invariance under the continuous group $SU(2) \times SU(2)$ [17, 18]. For future reference, let us explicitly work out these symmetries for the Hamiltonian (2.15).

The first $SU(2)$ corresponds to rotations in spin space, which can be interpreted also as a mixing of the $\sigma$ and $\tau$ operators. The generators are local in space if we express them using the original fermionic variables. However, when we work with the spin variables, the Jordan-Wigner strings appear. Formally we have

$$A_z = \sum_j \frac{\sigma^z - \tau^z}{2} \tag{2.21}$$

and

$$A_+ = \sum_j \left[ \left( \prod_{k<j} \sigma_k^z \tau_k^z \right) \sigma_j^+ \tau_j^- \right], \qquad A_- = \sum_j \left[ \left( \prod_{k<j} \sigma_k^z \tau_k^z \right) \sigma_j^- \tau_j^+ \right], \qquad (2.22)$$

that satisfy the standard $SU(2)$ algebra

$$[A_+, A_-] = A_z, \qquad [A_z, A_\pm] = \pm 2 A_\pm. \qquad (2.23)$$

For both periodic and open boundary conditions, the following condition holds

$$[A_z, H_1] = 0, \qquad (2.24)$$

while for the off-diagonal generators the symmetry relations

$$[A_\pm, H_1] = 0 \qquad (2.25)$$

hold only in the case of free boundary conditions, or formally in the infinite chain limit.

The second $SU(2)$ follows from the Shiba transformation. The idea is to perform a similarity transformation with either $\mathcal{S}^\sigma$ or $\mathcal{S}^\tau$, construct the $SU(2)$ generators of the modified Hamiltonian, and then to transform them back to the original $H_1$. In this way we obtain the $SU(2)$-generators (also called $\eta$-pairing generators)

$$B_z = \sum_j \frac{\sigma^z + \tau^z}{2}, \qquad (2.26)$$

and

$$B_+ = \sum_j \left[ \left( \prod_{k<j} \sigma_k^z \tau_k^z \right) \sigma_j^+ \tau_j^+ \right], \qquad B_- = \sum_j \left[ \left( \prod_{k<j} \sigma_k^z \tau_k^z \right) \sigma_j^- \tau_j^- \right]. \qquad (2.27)$$

Similarly to the $A$s, the operator $B_z$ commutes with $H_1$ (2.15) for both periodic and open boundary conditions and $B_\pm$ only in the open boundary case, or formally in the infinite volume limit.

## 3    Extension of the Hubbard model

We present a new integrable model of range 3 which is given by an extension of the Hubbard model, more precisely a deformation of the Hamiltonian $H_1$ (2.15).

**Definition**    The Hamiltonian is given by

$$H_3 = \sum_j \left[ h_{j,j+1}^\sigma + h_{j,j+1}^\tau + \frac{u}{4} l_{j,j+1,j+2}^\sigma l_{j,j+1,j+2}^\tau \right], \qquad (3.1)$$

where

$$l_{j,j+1,j+2}^\sigma = \sigma_{j+1}^z + \kappa \, (\sigma_j^x + \sigma_{j+2}^x) \sigma_{j+1}^x - \kappa^2 \, \sigma_j^x \sigma_{j+1}^z \sigma_{j+2}^x, \qquad (3.2)$$

and $l_{j,j+1,j+2}^\tau$ has the same expression but with the $\sigma$ matrices replaced by the $\tau$ matrices, and finally $h^\sigma$ and $h^\tau$ are given in (2.16). Sometimes we will omit the superscript and we will just use the notation $l_{j,j+1,j+2}$.

The Hamiltonian $H_3$ acts on the Hilbert space $\mathcal{V} = \otimes_{i=1}^L \mathcal{V}_i = \otimes_{i=1}^L (\mathbb{C}^2 \otimes \mathbb{C}^2)$ and the notation $h^\sigma$, $l^\sigma$ or $h^\tau$, $l^\tau$ identify respectively whether the operators appearing in $h$ and $l$ are respectively

$\sigma$ or $\tau$, so that if they act on the first or on the second copy of $\mathbb{C}^2$. As mentioned before, the notation $H_3$ signals that the density of the Hamiltonian acts on 3 sites of the spin chain, as it is clear from the subscript $_{j,j+1,j+2}$.

The parameters $u$ and $\kappa$ are the two independent coupling constants of the model; the model is Hermitian if they are both real. $u$ is the Hubbard interaction strength, while $\kappa$ is the deformation parameter. In this normalization, the original Hubbard model is restored for $\kappa = 0$. However, for $\kappa \neq 0$, there are two crucial differences:

1. The interaction term spans 3 consecutive sites.

2. Particle number conservation is broken.

It can be seen that the terms including the $\sigma^x$ and $\tau^x$ operators, that are linear or quadratic in the deformation parameter $\kappa$, manifestly break the $U(1)$ symmetries of the Hubbard model; they describe correlated particle creation and annihilation processes. In this respect, the model is analogous to the XYZ spin chain.

Given the many years of work that researchers spent with studying the Hubbard model and its generalizations one might wonder whether this model is indeed new or perhaps it exists in the literature. We performed an exhaustive search of the literature and did not find this model in any of its formulations (see also next Sections). All the previous extensions and deformations of the Hubbard model had two common properties [23, 25]:

1. The fundamental Hamiltonian was always nearest-neighbour interacting.

2. The model had (at least) two local $U(1)$ charges.

Our Hamiltonian (3.1) appears to differ from these properties, however, it could be that our $H_3$ is a rotated version of a linear combination of a two-site and three-site charge of a known model. In order to exclude this possibility we performed a search for a generic two-site charge $A$ which would commute with our $H_3$. Explicitly

$$[H_3, A] = [H_3, \sum_j a_{j,j+1}] = 0. \tag{3.3}$$

We used the program `Mathematica` [31] version 12.0 and found that, for generic coupling constants $u$ and $\kappa$, the only possibility for the operator density $a_{j,j+1}$ is to be of the form $a_{j,j+1} = b_j - b_{j+1} + \alpha \mathbb{1}$, which (after summation over $j$) lead to a trivial global charge. Thus our model *does not have any conserved charges with range less than three*. This excludes the possibility that our model is somehow included in the family of charges of a known model with a two site Hamiltonian.

**Integrability** The model given by $H_3$ is integrable: it has an infinite family of commuting local charges, which can be embedded into a transfer matrix construction. We checked this using the recently developed formalism of [32] for medium range spin chains and we explicitly found the $R$-matrix. For a brief review of the method see paragraph 5.2. Alternatively, we can treat the integrability properties by performing a duality transformation, see Section 4. In this way, the model becomes nearest-neighbour interacting and it allows for a more standard treatment.

Finally, we note a curious property of the three site interaction operator given in (3.2): for every $\kappa$ we have

$$(l_{j,j+1,j+2})^2 = (1 + \kappa^2)^2 \tag{3.4}$$

This property appears to follow from the integrability of the model and the structure of the Hamiltonian; we will discuss this relation in an upcoming publication [33].

We also note that the operators $l_{j,j+1,j+2}$ are non-commuting for generic values of $\kappa$:

$$[l_{j,j+1,j+2}(\kappa), l_{j,j+1,j+2}(\kappa')] = 0 \tag{3.5}$$

holds only if $\kappa = \kappa'$ (trivial) or if $\kappa\kappa' = -1$.

The special structure implies that $[l_{j,j+1,j+2}(\kappa), l_{j+2,j+3,j+4}(\kappa')] = 0$, while generally

$$[l_{j,j+1,j+2}(\kappa), l_{j+1,j+2,j+3}(\kappa')] \neq 0. \tag{3.6}$$

The latter commutation holds only in the case of the Hubbard model ($\kappa = \kappa' = 0$).

**Special points**  Apart from the point $\kappa = 0$, where the model becomes the Hubbard model, there are two more special points where the symmetry of the model is enhanced. The other special points of the model are at $\kappa = \pm 1$. In this case, (3.2) becomes

$$l_{j,j+1,j+2} = \pm(\sigma_j^x + \sigma_{j+2}^x)\sigma_{j+1}^x + \sigma_{j+1}^z \left(1 - \sigma_j^x \sigma_{j+2}^x\right). \tag{3.7}$$

This model possesses exactly two $U(1)$ charges,

$$Q_2^{\sigma_x} = \sum_j \sigma_j^x \sigma_{j+1}^x, \qquad\qquad Q_2^{\tau_x} = \sum_j \tau_j^x \tau_{j+1}^x. \tag{3.8}$$

In fact, it can be shown that $[Q_2^{\sigma_x}, H_3] = [Q_2^{\tau_x}, H_3] = 0$ if $H_3$ is computed from (3.1) with the three site interaction given by (3.7).

We proved this property by direct computation. Equivalently, it can be also easily checked after performing a duality transformation; this is presented in the next Section. We used the program Mathematica [31] version 12.0 to check that indeed these points are the only ones that admit a commuting charge which is at most of range 2.

# 4 The two-site model

Here we transform the previous model into a form where the Hamiltonian is two-site interacting. The transformation has its roots in the Kramers-Wannier duality [34]. Performing the duality transformation has advantages and disadvantages, which we will discuss.

## 4.1 The duality transformation – generalities

There are two ways to introduce the desired duality transformation: either via a real space description of the states, or formally on the level of the operators acting on the Hilbert space. We treat both formulations. In order to define the transformation, we need to consider the models with open boundary conditions.

For simplicity, let us consider just one copy of the local space $\mathbb{C}^2$, on which our previous $\sigma^a$ operators act. The same argument can be repeated for the $\tau^a$ operators acting on the second copy of $\mathbb{C}^2$. On the level of operators, the duality transformation is a particular Clifford transformation [35]: a mapping between operators with the following two requirements:

- Products of Pauli matrices are mapped to products of Pauli matrices (including possible multiplication with phases, but without producing linear combinations).

- The operator algebra is preserved.

The duality is then defined by the mapping

$$\sigma_j^z \to \sigma_{j-\frac{1}{2}}^x \sigma_{j+\frac{1}{2}}^x, \qquad\qquad \sigma_j^x \to \prod_{k=1}^{j} \sigma_{k-\frac{1}{2}}^z. \qquad (4.1)$$

Here we introduced half shifts for the space coordinates after the mapping; the physical meaning of these shifts is explained below.

We can use the operator algebra of the Pauli matrices to extend this mapping to all operators. For example a product of $\sigma^x$ operators is mapped to a single $\sigma^z$ matrix, in this way we obtain a symmetric formulation for the elementary steps:

$$\sigma_j^z \to \sigma_{j-\frac{1}{2}}^x \sigma_{j+\frac{1}{2}}^x, \qquad\qquad \sigma_j^x \sigma_{j+1}^x \to \sigma_{j+\frac{1}{2}}^z. \qquad (4.2)$$

The real space interpretation of this transformation is the following: working in the computational basis, we perform a rotation and afterwards we put spin-1/2 variables on the bonds between the original sites, such that the new variables measure the presence or the absence of a domain wall (kink or anti-kink). This is why we call these steps a "bond-site transformation".

To be more precise, let us assume that the model in question has spin reflection symmetry. Then we can map the Hilbert space of a chain of length $L$ to that of an other chain of length $L-1$, such that for each bond we put an up spin if the two neighbouring sites have the same orientation, and a down spin if they have different orientation. The original spin pattern can be reconstructed from the bonds up to a global spin reflection step, which preserves all values of the bonds[1]. Denoting the new variables with space positions at half shifts, the mapping on the operatorial level becomes simply

$$\sigma_j^z \sigma_{j+1}^z \to \sigma_{j+\frac{1}{2}}^z. \qquad (4.3)$$

A single spin flip on the original chain necessarily changes the values on two bonds, thus we obtain the other elementary transformation rule

$$\sigma_j^x \to \sigma_{j-\frac{1}{2}}^x \sigma_{j+\frac{1}{2}}^x. \qquad (4.4)$$

These are not yet identical to the steps (4.1)-(4.2). In order to achieve the same formulas, one needs to perform a global rotation *before the bond site transformation*, which maps

$$\sigma^z \to \sigma^x, \qquad\qquad \sigma^x \to \sigma^z, \qquad\qquad \sigma^y \to -\sigma^y. \qquad (4.5)$$

Combining this rotation with (4.3)-(4.4) we obtain the transformation rules (4.1)-(4.2).

The advantage of using the formulas (4.1)-(4.2) is that they describe an involutive transformation, so that applying the transformations twice will produce the initial model.

This bond-site transformation has its origin in the Kramers-Wannier duality, which can be used to determine the critical point of the Ising model on the square lattice. It can also be applied to the 1D quantum Ising chain, where it acts as a self-duality [34]. More recently the same transformation was also used in the "folded XXZ model" [36, 37].

The transformation is non-local: a subset of local operators remains local after the mapping, but the remaining subset (including $\sigma^x$ by the definition (4.1)) becomes truly non-local. In those cases when the local Hamiltonian density is mapped to local operators it is possible to define the bond-site transformed model also with periodic boundary conditions. However, in this case the two models are strictly speaking not equivalent. This can be seen on the level of the real space

---

[1]We remark that if the model does not have spin reflection, those statements remain true with the addition that we need to know the state of the first site. Furthermore, in this case the Hamiltonian becomes non-local.

transformation: in the periodic case any state has an even number of domain walls, therefore it is mapped to a state with an even number of down spins. Therefore, the sectors of the new model with odd down spins do not correspond to the states of the original model. This difference should not affect the thermodynamic properties of the models, but it is crucial for the comparison of finite volume quantities.

## 4.2   Model with nearest-neighbour interactions

Now we compute the transformation of our model Hamiltonian $H_3$ (3.1). We perform the bond-site transformation for the $\sigma$ and $\tau$ matrices as well.

First we transform the kinetic terms. They are odd with respect to spin reflection, however, this does not cause any complications. Starting with the $\sigma$ matrices, we use the rewriting

$$\sigma_j^y \sigma_{j+1}^x - \sigma_j^x \sigma_{j+1}^y = i(\sigma_{j+1}^z - \sigma_j^z)\sigma_j^x \sigma_{j+1}^x \qquad \rightarrow$$
$$i(\sigma_{j+\frac{1}{2}}^x \sigma_{j+\frac{3}{2}}^x - \sigma_{j-\frac{1}{2}}^x \sigma_{j+\frac{1}{2}}^x)\sigma_{j+\frac{1}{2}}^z = \sigma_{j+\frac{1}{2}}^y \sigma_{j+\frac{3}{2}}^x - \sigma_{j-\frac{1}{2}}^x \sigma_{j+\frac{1}{2}}^y. \tag{4.6}$$

We see that after transformation, the kinetic term is now localized on three sites. However, summing over these contributions on an infinite chain (or extending the transformation formally to periodic boundary conditions) we see that *the integrated kinetic term is self-dual*. This means that for these particular models the bond-site transformation will only change the interaction terms.

Let us now perform the transformation for the total Hamiltonian $H_3$ of (3.1). Now it is more convenient to use a different parametrization. We introduce the coupling constants $U$ and $\theta$ such that

$$\kappa = \tan\frac{\theta}{2}, \qquad\qquad u = 8U \sec^4\frac{\theta}{2}. \tag{4.7}$$

Direct computation then gives[2]

$$H_2 = \sum_j \left[h_{j,j+1}^\sigma + h_{j,j+1}^\tau + 2U\, L_{j,j+1}^\sigma L_{j,j+1}^\tau\right], \tag{4.8}$$

where

$$L_{j,j+1}^\sigma = \frac{\sin\theta}{2}(\sigma_j^z + \sigma_{j+1}^z) + \cos\theta(\sigma_j^- \sigma_{j+1}^- + \sigma_j^+ \sigma_{j+1}^+) + (\sigma_j^- \sigma_{j+1}^+ + \sigma_j^+ \sigma_{j+1}^-), \tag{4.9}$$

the kinetic terms are given in (2.16). The notation $H_2$ signals that the interaction term is acting on 2 sites of the chain. The parameters $U$ and $\theta$ are two coupling constants and $H_2$ is Hermitian if both are real. For $\theta$ we choose the fundamental domain $[-\pi, \pi]$.

In a concrete matrix representation we can write

$$L_{j,j+1} = \begin{pmatrix} \sin\theta & 0 & 0 & \cos\theta \\ 0 & 0 & 1 & 0 \\ 0 & 1 & 0 & 0 \\ \cos\theta & 0 & 0 & -\sin\theta \end{pmatrix}. \tag{4.10}$$

This matrix is of 8-vertex type [38]: it does not conserve the $S^z$ particle numbers, but particle creation and annihilation only happen in pairs. The structure of the resulting Hamiltonian $H_2$ is

---

[2]We remark that the letter $L$ was also used for the number of sites of the spin chain. The context will clearly tell how to distinguish the two cases.

the same as in the Hubbard model and its various extensions, see for example [21, 27]. However, now the interaction $L_{j,j+1}$ does not conserve particle number for a generic $\theta$. After investigating the special points, we will establish that (contrary to the three-site model given by $H_3$) the family of Hamiltonians (4.8) *does not include the actual Hubbard model* for any choice of $\theta$.

The curious property[3] (3.4) holds also after the bond-site transformation: direct computation confirms that

$$(L_{j,j+1})^2 = 1. \tag{4.11}$$

We also note that the matrix $L_{j,j+1}$ is free fermionic, which is evident from the representation (4.9): performing again a Jordan-Wigner transformation we find terms which are only bilinear in the fermionic operators. In fact, the models obtained by the Hamiltonian $\sum_j L_{j,j+1}$ are known in the literature as the XYh models [39]. However, our model involves the coupling of $L^\sigma_{j,j+1}$ and $L^\tau_{j,j+1}$, therefore it is interacting.

In parallel with the non-commutativity (3.6), we observe that for a generic value of $\theta$

$$[L_{j,j+1}, L_{j+1,j+2}] \neq 0. \tag{4.12}$$

Let us now discuss the symmetries of (4.8) for a generic value of $\theta$. First of all, we do not find any continuous symmetries. However, there are discrete symmetries. In particular, the Shiba transformations (2.18) preserve the kinetic terms and both of them negate the sign of the coupling constant $U$. Therefore their combination is a discrete symmetry:

$$\mathcal{S}^\tau \mathcal{S}^\sigma H_2 \mathcal{S}^\sigma \mathcal{S}^\tau = H_2. \tag{4.13}$$

Because both interaction matrices create/annihilate particles in pairs, the "fermionic parity" is conserved for both sub-chains:

$$[Z_\sigma, H_2] = [Z_\tau, H_2] = 0 \tag{4.14}$$

where

$$Z_\sigma = \prod_{j=1}^{L} \sigma_j^z, \qquad\qquad Z_\tau = \prod_{j=1}^{L} \tau_j^z. \tag{4.15}$$

This property also holds for the range 3 spin chain (3.1).

## 4.3 Special points

Just as in the previous case, there are some points where the the family given by $H_2$ (4.8) has additional symmetries, those are $\theta = 0, \pm\pi/2$.

**The choice $\theta = 0$.** In this point, the interaction operator $L_{j,j+1}$ (4.9) is

$$L_{j,j+1} = \sigma_j^x \sigma_{j+1}^x \tag{4.16}$$

which is represented by an anti-diagonal matrix. This particular model is *the bond-site transformation of the Hubbard model*. Accordingly, it possesses two $U(1)$-charges given by $Q_2^{\sigma^x}$ and $Q_2^{\tau^x}$ defined in (3.8), which can be extended to two $SU(2)$ algebras.

The known coordinate Bethe Ansatz solution of the Hubbard model [1] can be used to construct eigenstates of the model (4.8) with $L_{j,j+1}$ given in (4.16). The idea is to perform

---

[3]Note that (3.4) can be also made equal to 1 by renormalizing the $l$ operator.

the bond-site transformation on the level of the eigenstates. However, as remarked earlier, this computation will only produce those states which have an even number of down spins for both the $\sigma$ and the $\tau$ sub-lattices. It follows from the commutation relations (4.14) that this "parity" is indeed consistent with the Hamiltonian. At present it is not known how to treat the odd sub-sectors.

**The choice $\theta = \pm\pi/2$.** In this case we obtain the bond-site transformation of the model given by (3.7). For $\theta = \pi/2$ we find the Hamiltonian (4.8) with the interaction matrix

$$L_{j,j+1} = \begin{pmatrix} 1 & & & \\ & & 1 & \\ & 1 & & \\ & & & -1 \end{pmatrix} = \sigma_j^+\sigma_{j+1}^- + \sigma_j^-\sigma_{j+1}^+ + \frac{1}{2}\left(\sigma_i^z + \sigma_{i+1}^z\right). \tag{4.17}$$

The case with $\theta = -\pi/2$ is not independent from the one just shown: one can apply a unitary off-diagonal local basis transformation and a re-definition of the coupling constant $U$ to relate the two models.

These cases are special because they enjoy two $U(1)$-symmetries due to the particle conservation: the Hamiltonian now commutes with $N$ and $S_z$ given by (2.4). More generally, it formally commutes with the all the generators (2.21) up to boundary terms.

Interestingly, this model can be obtained as a particular limit of a known extension of the Hubbard model [21], which originates from the $R$-matrix of Shastry. In the model of [21] the Hamiltonian is

$$H_{j,j+1} = \sigma_j^x\sigma_{j+1}^x + \sigma_j^y\sigma_{j+1}^y + \tau_j^x\tau_{j+1}^x + \tau_j^y\tau_{j+1}^y + \alpha L_{j,j+1}^\sigma L_{j,j+1}^\tau \tag{4.18}$$

with the two-site interaction matrix given by

$$L_{j,j+1} = \begin{pmatrix} \cos(2v) & 0 & 0 & 0 \\ 0 & 1 & -\sin(2v) & 0 \\ 0 & \sin(2v) & -1 & 0 \\ 0 & 0 & 0 & -\cos(2v) \end{pmatrix}, \tag{4.19}$$

with $\alpha, v$ two independent parameters. The case $v = 0$ is the Hubbard model (up to a trivial global shift of $H$). Let us now perform the similarity transformation with the diagonal operator (2.12) and change the normalization by a factor of $1/2$. Then we obtain

$$\frac{1}{2}D^{-1}HD = \sum_j \left[h_{j,j+1}^\sigma + h_{j,j+1}^\tau + \alpha L_{j,j+1}^\sigma L_{j,j+1}^\tau\right] \tag{4.20}$$

where now

$$L_{j,j+1} = \begin{pmatrix} \cos(2v) & 0 & 0 & 0 \\ 0 & 1 & -i\sin(2v) & 0 \\ 0 & -i\sin(2v) & -1 & 0 \\ 0 & 0 & 0 & -\cos(2v) \end{pmatrix} \tag{4.21}$$

and $h^\sigma$ is given in (2.17).

We can now take in (4.21) the limit $v \to i\infty$ to get the $L_{j,j+1}$ in (4.17) and $\alpha \to 2U$ will make the total $H$ equivalent to $H_2$.

It is remarkable that two special points of the model of [21] are reproduced by two very different versions of our models: the actual Hubbard model ($v = 0$ of (4.18)) is found as a special point of our three-site Hamiltonian $H_3$, whereas the $v \to i\infty$ limit of (4.20) can be found in our two-site family $H_2$. Perhaps there is a larger family of integrable models which contains all these special points.

# 5 Integrability

In this Section we rigorously prove the integrability of our models by embedding them into the Quantum Inverse Scattering Approach [40], which is the canonical framework to treat integrable spin chains. As a by-product, we find a solution of the famous Yang-Baxter relations, which has some unusual spectral parameter dependence and appears to be new.

First we consider the two-site model, and afterwards we turn to the three-site model.

## 5.1 Two-site model

Here we treat the integrability of the Hamiltonian (4.8) for an arbitrary value of the coupling constants $U$, $\theta$. It is our goal to construct families of commuting transfer matrices, which generate a set of commuting local charges for each value of $U$ and $\theta$.

Consider a Lax operator $\mathcal{L}_{aj}(u, \mu)$ which acts on the tensor product of an auxiliary space $V_a \approx \mathbb{C}^4$ and a local 4-dimensional space, having two complex valued "spectral parameters" $u$ and $\mu$. The transfer matrix $t(u|\mu)$ is a matrix product operator (MPO) defined as the trace

$$t(u|\mu) = \text{Tr}_a \left[ \mathcal{L}_{aL}(u, \mu) \dots \mathcal{L}_{a2}(u, \mu) \mathcal{L}_{a1}(u, \mu). \right] \tag{5.1}$$

The transfer matrices form a commuting family for fixed $\mu$:

$$[t(u_1|\mu), t(u_2|\mu)] = 0, \tag{5.2}$$

if the Lax operators satisfy the intertwining relation

$$R_{ab}(u, v) \mathcal{L}_{an}(u, \mu) \mathcal{L}_{bn}(v, \mu) = \mathcal{L}_{bn}(v, \mu) \mathcal{L}_{an}(u, \mu) R_{ab}(u, v). \tag{5.3}$$

Consistency of the intertwining relations imply that the $R$-matrix should satisfy the quantum Yang-Baxter equation

$$R_{12}(u_1, u_2) R_{13}(u_1, u_3) R_{23}(u_2, u_3) = R_{23}(u_2, u_3) R_{13}(u_1, u_3) R_{12}(u_1, u_2). \tag{5.4}$$

If the $R$-matrix is regular, *i.e.* $R(u, u) = P$, where $P$ is the permutation operator, then the logarithmic derivative of the transfer matrix at $u = \mu$ defines a Hamiltonian with nearest-neighbour interactions:

$$H_2(\mu) = \left. \frac{d}{du} \log t(u|\mu) \right|_{u=\mu} \tag{5.5}$$

and for the other charges

$$\mathbb{Q}_{r+1}(\mu) \sim \left. \frac{d^r}{du^r} t(u|\mu) \right|_{u=\mu}. \tag{5.6}$$

The higher logarithmic derivatives define the higher conserved charges that characterise an integrable model. From (5.2) and (5.6) one gets

$$[\mathbb{Q}_r(\mu), \mathbb{Q}_s(\mu)] = 0. \tag{5.7}$$

Our explicit solution for the $R$-matrix is presented in Appendix A. We reproduce the correct Hamiltonian (4.8) if we take our Lax operator to be related to the $R$-matrix in the following way

$$\mathcal{L}(u, \mu) \equiv R(\alpha \, u, \mu), \tag{5.8}$$

where

$$\alpha = \frac{2}{\mathrm{dn}\,(\mu\,|k^2)}, \qquad \mu = \mathrm{cn}^{-1}\left(-\sec\theta\,|k^2\right), \qquad k = \frac{2iU\cos^2\theta}{\sqrt{1 + U^2\sin^2(2\theta)}}. \qquad (5.9)$$

We have checked that the $R$-matrix from Appendix A satisfies the Yang-Baxter equation and braiding unitarity. We would also like to point out that this $R$-matrix has a very non-trivial functional dependence on the spectral parameter. First, the $R$-matrix is of non-difference form. This can be easily seen since it depends both on sums and differences of Jacobi elliptic functions. Second, it cannot be completely expressed in terms of the usual Jacobi elliptic functions, due to terms of the form

$$\sin\frac{1}{2}\Big[\mathrm{am}(u|k^2) - \mathrm{am}(v|k^2)\Big], \qquad\qquad \sec\frac{1}{2}\Big[\mathrm{am}(u|k^2) - \mathrm{am}(v|k^2)\Big], \qquad (5.10)$$

where am is the Jacobi amplitude. This can only be expressed in the Jacobi elliptic functions sn, cn, dn by introducing square roots. To the best of our knowledge, this $R$-matrix is new and we have also not encountered a model with this functional dependence before.

## 5.2  $R$-matrix for the 3-site model

The integrability of the two-site version also implies that our original three-site formulation $H_3$ is integrable, in the sense that it also possesses an infinite family of local conserved charges. This can be proven in two ways.

First of all, we can show that the higher charges of the two-site Hamiltonian $H_2$ remain all local if we perform the duality transformation to the three-site family. This follows from the fact that $H_2$ commutes with $Z^\sigma$ and $Z^\tau$, therefore it can contain an even number of Pauli matrices which cause a spin flip. The duality transformation (4.2) produces non-local operators only for an odd number of spin flipping Pauli matrices. It follows that all charges of the two site models remain local after the transformation.

Alternatively, we also applied the formalism of [32] to directly prove integrability of the three site model by constructing a Lax operator with a higher dimensional auxiliary space. This auxiliary space is typically a tensor product of copies of the elemenatry spaces.

For an integrable spin chain with three site interactions the auxiliary space is a tensor product of two copies of the fundamental vector space. Therefore, the Lax matrix is an operator which acts on three spaces, one physical space and two auxiliary spaces. It is denoted as $\mathcal{L}_{a,b,j}(u)$, where $a$ and $b$ are the two auxiliary spaces, and $j$ refers to a physical space. The transfer matrix is defined as

$$t(u) = \mathrm{Tr}_a\left[\mathcal{L}_{a,b,L}(u)\ldots\mathcal{L}_{a,b,2}(u)\mathcal{L}_{a,b,1}(u).\right] \qquad (5.11)$$

As for the two site model, the conserved charges are defined by taking the logarithmic derivative of the transfer matrix.

Following [32] we also introduce

$$\check{\mathcal{L}}(u) = P_{a,j}P_{b,j}\mathcal{L}_{a,b,j}(u), \qquad (5.12)$$

where $P$ stands again for the permutation operator.

The transfer matrices form a commuting family, which is established from the fundamental intertwining relation:

$$\check{R}_{23,45}(u_1,u_2)\check{\mathcal{L}}_{123}(u_1)\check{\mathcal{L}}_{345}(u_2) = \check{\mathcal{L}}_{123}(u_2)\check{\mathcal{L}}_{345}(u_1)\check{R}_{12,34}(u_1,u_2). \qquad (5.13)$$

Here $R(u,v)$ is the $R$-matrix, which depends on two spectral parameters, and it acts on a four-fold tensor product space.

Since the 2-site model Hamiltonian $H_2$ in (4.8) is related to the 3-site one (3.1) by a bond-site transformation, it is reasonable to assume that the $\mathcal{L}$ matrix is given by the bond site transformed version of the 2-site one. The necessary steps for the bond-site transformed models can be extracted from Section V.A of [32], with the only difference that here the Lax operator will also depend on two spectral parameters (as will be clarified in the following). For completeness we present here the details of this procedure.

The starting point is the $R$-matrix for the 2 site model given in appendix A, in particular we work with

$$\check{R}_{a,b}(u,v) = P_{a,b}R_{a,b}(u,v), \tag{5.14}$$

We first perform the rotation (4.5) followed by the transformations (4.3)-(4.4). The result will be a range-three operator that we will identify as the Lax-matrix. Accordingly, in this case the Lax matrix will have two spectral parameters:

$$\check{R}_{j,j+1}(u,v) \quad \rightarrow \quad \check{\mathcal{L}}_{j,j+1,j+2}(u,v), \tag{5.15}$$

and the intertwining relation becomes

$$\check{R}_{23,45}(u_1,u_2)\check{\mathcal{L}}_{123}(u_1,u_3)\check{\mathcal{L}}_{345}(u_2,u_3) = \check{\mathcal{L}}_{123}(u_2,u_3)\check{\mathcal{L}}_{345}(u_1,u_3)\check{R}_{12,34}(u_1,u_2). \tag{5.16}$$

In all of the computations below the second spectral parameter of the Lax operator is seen as an outer (spectator) parameter, for which we do not introduce intertwining relations.

It follows from the bond-site transformation that

$$[\check{\mathcal{L}}_{123}(u_1,u_3), \check{\mathcal{L}}_{345}(u_2,u_3)] = 0. \tag{5.17}$$

We can multiply equation (5.16) from the left by $\check{\mathcal{L}}_{345}^{-1}(u_2,u_3)$ and the right by $\check{\mathcal{L}}_{123}^{-1}(u_1,u_3)$ and using the properties (5.17) we get

$$\check{\mathcal{L}}_{345}^{-1}(u_1,u_3)\check{R}_{23,45}(u_1,u_2)\check{\mathcal{L}}_{345}(u_2,u_3) = \check{\mathcal{L}}_{123}(u_2,u_3)\check{R}_{12,34}(u_1,u_2)\check{\mathcal{L}}_{123}^{-1}(u_1,u_3) \tag{5.18}$$

and we can observe that the l.h.s. acts trivially in the space 5 and the r.h.s. acts trivially in the space 1, so they have to be equal to a three site operator $\mathcal{G}_{234}(u_1,u_2,u_3)$.

Explicitly, we got

$$\check{\mathcal{G}}_{234}(u_1,u_2,u_3) = \check{\mathcal{L}}_{345}^{-1}(u_1,u_3)\check{R}_{23,45}(u_1,u_2)\check{\mathcal{L}}_{345}(u_2,u_3) \tag{5.19}$$

and

$$\check{\mathcal{G}}_{234}(u_1,u_2,u_3) = \check{\mathcal{L}}_{123}(u_2,u_3)\check{R}_{12,34}(u_1,u_2)\check{\mathcal{L}}_{123}^{-1}(u_1,u_3). \tag{5.20}$$

And for the $R$ matrix we obtain the expressions

$$\check{R}_{12,34}(u_1,u_2) = \check{\mathcal{L}}_{123}^{-1}(u_2,u_3)\check{\mathcal{G}}_{234}(u_1,u_2,u_3)\check{\mathcal{L}}_{123}(u_1,u_3) \tag{5.21}$$

and

$$\check{R}_{23,45}(u_1,u_2) = \check{\mathcal{L}}_{345}(u_1,u_3)\check{\mathcal{G}}_{234}(u_1,u_2,u_3)\check{\mathcal{L}}_{345}^{-1}(u_2,u_3). \tag{5.22}$$

Direct computation confirms that the dependence on $u_3$ drops out.

$\mathcal{G}$ is also a range-three operator, that a-priori can depends in all the spectral parameter $u_1, u_2, u_3$. By direct computation, the dependence on the third spectral parameter drops out and $\check{\mathcal{G}}_{123}(u_1,u_2,u_3) = \check{\mathcal{G}}_{123}(u_1,u_2)$. It turns out that a solution to the Yang-Baxter equations (5.16) is found if we choose the Lax operator to be equal to $\check{\mathcal{G}}_{123}(u_1,u_2)$. Furthermore, the $R$-matrix is then written as

$$\check{R}_{12,34}(u_1,u_2) = \check{\mathcal{L}}_{123}^{-1}(u_2,u_3)\check{\mathcal{L}}_{234}(u_1,u_3)\check{\mathcal{L}}_{123}(u_1,u_3) \tag{5.23}$$

and

$$\check{R}_{23,45}(u_1, u_2) = \check{\mathcal{L}}_{345}(u_1, u_3)\check{\mathcal{L}}_{234}(u_1, u_3)\check{\mathcal{L}}_{345}^{-1}(u_2, u_3). \tag{5.24}$$

To summarize, we assumed that the $\mathcal{G}$ operator is the Lax matrix of the three site model and the latter is the bond site transformed version of the 2 site $R$ matrix. To check the validity of these assumptions, using the program `Mathematica` [31] version 12.0, we checked the consistency relation between (5.21) and (5.22), that is

$$\check{\mathcal{L}}_{123}(u_2, u_3)\check{\mathcal{L}}_{234}(u_1, u_3)\check{\mathcal{G}}_{123}(u_1, u_2) = \check{\mathcal{G}}_{234}(u_1, u_2)\check{\mathcal{L}}_{123}(u_1, u_3)\check{\mathcal{L}}_{234}(u_2, u_3). \tag{5.25}$$

or equivalently of (5.23) and (5.24),

$$\check{\mathcal{L}}_{123}(u_2, u_3)\check{\mathcal{L}}_{234}(u_1, u_3)\check{\mathcal{L}}_{123}(u_1, u_2) = \check{\mathcal{L}}_{234}(u_1, u_2)\check{\mathcal{L}}_{123}(u_1, u_3)\check{\mathcal{L}}_{234}(u_2, u_3). \tag{5.26}$$

It follows that the $\check{R}$ matrix obtained from either (5.21) or (5.22) satisfy the following YBE

$$\check{R}_{34,56}(u_1, u_2)\check{R}_{12,34}(u_1, u_3)\check{R}_{34,56}(u_2, u_3) = \check{R}_{12,34}(u_2, u_3)\check{R}_{34,56}(u_1, u_3)\check{R}_{12,34}(u_1, u_2). \tag{5.27}$$

## 6 Large coupling limits

Here we investigate the large coupling limits of the models. The idea is to take the limit $u \to \infty$ and $U \to \infty$ of the Hamiltonians (3.1) and (4.8). In this limit the interaction term between the two sub-chains will dominate, which is equivalent to setting the kinetic terms equal to zero.

For a generic coupling $\kappa$ and $\theta$ the non-commutativity in (3.6) and (4.12) imply that the interaction terms generate dynamics in the system. This means that non-trivial integrable models are obtained by the direct $u, U \to \infty$ limit and a simple rescaling. In this way we obtain the models

$$H_3^\infty = \sum_j l_{j,j+1,j+2}^\sigma l_{j,j+1,j+2}^\tau, \qquad H_2^\infty = \sum_j L_{j,j+1}^\sigma L_{j,j+1}^\tau, \tag{6.1}$$

with $l_{j,j+1,j+2}$ and $L_{j,j+1}$ given by (3.2) and (4.10), respectively.

The two models are the bond-site transformations of each other. To our best knowledge, these models are also new. Their integrability follows directly from the constructions of the $R$-matrices for the general cases.

In the case of the two site model the Hamiltonian $H_2^\infty$ given in (6.1) is obtained via the substitutions

$$\mathcal{L}(u, \mu) \equiv R(\alpha u, \mu), \tag{6.2}$$

$$\alpha = \frac{2i \cos^2 \theta}{k}, \qquad\qquad \mu = \frac{1}{k}K\left(\frac{1}{k^2}\right), \qquad\qquad k = i \cot \theta, \tag{6.3}$$

where $K$ is the elliptic integral of the first kind[4]. For the three site model, the $R$-matrix can be obtained in the same way as explained above at the end of Section 5.

---

[4]In order to get this result, we used the relations of [41],

$$\mathrm{dn}\left(v \,|\, k^2\right) = \mathrm{cn}\left(v\,k \,\Big|\, \frac{1}{k^2}\right), \qquad \mathrm{cn}\left(v \,|\, k^2\right) = \mathrm{dn}\left(v\,k \,\Big|\, \frac{1}{k^2}\right), \qquad \mathrm{sn}\left(v \,|\, k^2\right) = \frac{1}{k}\mathrm{sn}\left(v\,k \,\Big|\, \frac{1}{k^2}\right), \tag{6.4}$$

and we chose the branch cut $\sqrt{\sec^2 \theta} \cos \theta = -1$.

The situation is somewhat different in the case of the Hubbard model, corresponding to $\kappa = 0$ and its bond-site transformed model ($\theta = 0$). In this case both $H_3^\infty$ and $H_2^\infty$ become a sum of commuting operators:

$$H_3^\infty \to \sum_j \sigma_j^z \tau_j^z, \qquad H_2^\infty \to \sum_j \sigma_j^x \sigma_{j+1}^x \tau_j^x \tau_{j+1}^x. \tag{6.5}$$

These operators do not generate non-trivial dynamics. However, it is still meaningful to investigate the $U \to \infty$ limit of the Hubbard model.

In this limit the double occupancies become forbidden and one obtains the so-called $t - 0$ model as an effective theory, see for example [2, 42]. We do not discuss this limit further in this work.

## 7  Conclusions and Outlook

In this work we introduced different generalizations of the Hubbard model. The model Hamiltonian (3.1) is a generalization with three-site interactions, such that the special choice $\kappa = 0$ reproduces the original Hubbard Hamiltonian. In contrast, the two-site formulation (4.8) has a Hamiltonian with similar structure, but this family does not include the Hubbard model itself. The two distinguished properties of the Hamiltonian (3.1) are that it is three-site interacting and it breaks the $U(1)$-symmetries of the original model. Correspondingly, the $R$-matrix of the model involves an unusual dependence on the elliptic functions (similar to the case of the XYZ spin chain), and it appears to be new.

The three site interaction with a tunable deformation parameter is interesting on its own right. Most integrable models in the literature either have nearest-neighbour interactions, or true long range interactions with some coupling/deformation parameters. The recent work [32] set up a framework to study and classify models with medium range interactions: in these cases the Hamiltonian density has a finite range bigger than two. None of the the examples found in [32] is a continuous deformation of a nearest neighbour interacting model. In this sense our model is unique. The Bariev model [43] is somewhat similar, because in that case the coupling can be tuned such that the model falls apart into two disconnected XX chains with nearest-neighbour coupling. However, this situation is different, because the "deformation" parameter couples two chains and it does not modify a single nearest-neighbour interacting model. We stress that in our case the undeformed model with $\kappa = 0$ is still interacting, it is given by the Hubbard model with a non-zero coupling.

Even though we could clarify the integrability structure of our models, we leave the actual solution (construction of eigenstates) to further work. The breaking of the $U(1)$ symmetries makes the problem considerably more complicated than in the case of the Hubbard model. We expect that some combination of the nested Bethe Ansatz with methods used to solve the XYZ spin chain needs to be used.

**Acknowledgments**

We are thankful to Frank Göhmann and Eric Ragoucy for useful discussions. We would like to thank Tamás Gombor and Ana L. Retore for valuable comments on the manuscript and for interesting discussions. MdL was supported by SFI, the Royal Society and the EPSRC for funding under grants UF160578, RGF\R1\181011, RGF\EA\180167 and 18/EPSRC/3590. CP was supported by the grant RGF\R1\181011. This research was supported in part by the National Science Foundation under Grant No. NSF PHY-1748958.

# A  $R$-matrix

Here we publish the concrete $R$-matrix, which describes the integrability properties of the two-site Hamiltonian (4.8). Upon request we are happy to share a Mathematica notebook containing this $R$-matrix.

The actual matrix form reads

$$R = \begin{pmatrix} r_8 & 0 & 0 & 0 & 0 & -r_{12} & 0 & 0 & 0 & 0 & -r_{12} & 0 & 0 & 0 & 0 & r_1 \\ 0 & r_6 & 0 & 0 & r_7 & 0 & 0 & 0 & 0 & 0 & 0 & r_{11} & 0 & 0 & 0 & 0 \\ 0 & 0 & r_6 & 0 & 0 & 0 & 0 & r_{11} & r_7 & 0 & 0 & 0 & 0 & 0 & 0 & 0 \\ 0 & 0 & 0 & r_2 & 0 & 0 & -r_9 & 0 & 0 & -r_9 & 0 & 0 & r_4 & 0 & 0 & 0 \\ 0 & r_7 & 0 & 0 & -r_5 & 0 & 0 & 0 & 0 & 0 & 0 & 0 & 0 & 0 & r_{11} & 0 \\ -r_{12} & 0 & 0 & 0 & 0 & r_{10} & 0 & 0 & 0 & 0 & r_1 & 0 & 0 & 0 & 0 & r_{12} \\ 0 & 0 & 0 & -r_9 & 0 & 0 & r_3 & 0 & 0 & r_4 & 0 & 0 & r_9 & 0 & 0 & 0 \\ 0 & 0 & r_{11} & 0 & 0 & 0 & 0 & r_5 & 0 & 0 & 0 & 0 & 0 & r_7 & 0 & 0 \\ 0 & 0 & r_7 & 0 & 0 & 0 & 0 & 0 & -r_5 & 0 & 0 & 0 & 0 & r_{11} & 0 & 0 \\ 0 & 0 & 0 & -r_9 & 0 & 0 & r_4 & 0 & 0 & r_3 & 0 & 0 & r_9 & 0 & 0 & 0 \\ -r_{12} & 0 & 0 & 0 & 0 & r_1 & 0 & 0 & 0 & 0 & r_{10} & 0 & 0 & 0 & 0 & r_{12} \\ 0 & r_{11} & 0 & 0 & 0 & 0 & 0 & 0 & 0 & 0 & 0 & r_5 & 0 & 0 & r_7 & 0 \\ 0 & 0 & 0 & r_4 & 0 & 0 & r_9 & 0 & 0 & r_9 & 0 & 0 & r_2 & 0 & 0 & 0 \\ 0 & 0 & 0 & 0 & 0 & 0 & 0 & r_7 & r_{11} & 0 & 0 & 0 & 0 & -r_6 & 0 & 0 \\ 0 & 0 & 0 & 0 & r_{11} & 0 & 0 & 0 & 0 & 0 & 0 & r_7 & 0 & 0 & -r_6 & 0 \\ r_1 & 0 & 0 & 0 & 0 & r_{12} & 0 & 0 & 0 & 0 & r_{12} & 0 & 0 & 0 & 0 & r_8 \end{pmatrix} ,$$

(A.1)

where we suppressed the dependence on two spectral parameters, i.e. $r_i = r_i(u,v)$.

The matrix elements are

$$r_1 = -\frac{2ik g_{u,v}}{\mathrm{dn}_u + \mathrm{dn}_v}, \qquad r_4 = f_{u,v}, \qquad r_7 = 1,$$

$$i\, r_9 = \frac{\mathrm{cn}_u - \mathrm{cn}_v}{\mathrm{sn}_u + \mathrm{sn}_v}, \qquad i\,k\,r_{11} = \frac{\mathrm{dn}_u - \mathrm{dn}_v}{\mathrm{sn}_u + \mathrm{sn}_v}, \qquad \frac{r_{12}}{k} = \frac{\mathrm{cn}_u - \mathrm{cn}_v}{\mathrm{dn}_u + \mathrm{dn}_v},$$

$$r_5 + r_6 = -2i g_{u,v}, \qquad k\,(r_5 - r_6) = (\mathrm{dn}_v - \mathrm{dn}_u) f_{u,v},$$

$$r_3 + r_{10} + r_8 - r_2 = 2f_{u,v}, \qquad r_3 + r_{10} + r_2 - r_8 = \frac{4\,i\,k(\mathrm{cn}_u - \mathrm{cn}_v)}{(\mathrm{dn}_u + \mathrm{dn}_v)(\mathrm{sn}_u + \mathrm{sn}_v) f_{u,v}},$$

$$\frac{r_{10}(\mathrm{dn}_u + i\,k\,\mathrm{cn}_u\mathrm{sn}_u) + r_8(\mathrm{dn}_u - i\,k\,\mathrm{cn}_u\mathrm{sn}_u)}{r_4} = \mathrm{sn}_u^2(\mathrm{dn}_v - \mathrm{dn}_u) + \frac{2\mathrm{dn}_u}{f_{u,v}^2} + \frac{2k^2\mathrm{sn}_u(\mathrm{cn}_u - \mathrm{cn}_u) g_{u,v}}{(\mathrm{dn}_u + \mathrm{dn}_v) f_{u,v}},$$

$$\frac{r_3(\mathrm{dn}_u + i\,k\,\mathrm{cn}_u\mathrm{sn}_u) + r_2(\mathrm{dn}_u - i\,k\,\mathrm{cn}_u\mathrm{sn}_u)}{r_4} =$$
$$r_{12}\left(\frac{4\mathrm{dn}_u}{(\mathrm{sn}_u + \mathrm{sn}_v) f_{u,v}^2} + \mathrm{sn}_u (\mathrm{dn}_v - \mathrm{dn}_u)\right) + \frac{f_{u,v} g_{u,v}}{k}\left(\frac{2k^2\mathrm{sn}_u^2}{f_{u,v}^2} - \mathrm{dn}_u\mathrm{dn}_v + \mathrm{dn}_u^2\right),$$

where we defined the shorthand notations for the Jacobi functions:

$$\mathrm{cn}_u = \mathrm{cn}\left(u|k^2\right), \qquad \mathrm{sn}_u = \mathrm{sn}\left(u|k^2\right), \qquad \mathrm{Am}_u = \mathrm{am}\left(u|k^2\right), \qquad \text{(A.2)}$$
$$\mathrm{dn}_u = \mathrm{dn}\left(u|k^2\right), \qquad\qquad\qquad\qquad \text{(A.3)}$$

$k$ and the spectral parameters are related to $U$, $\theta$ by (5.9) and we defined for simplicity

$$g_{u,v} = \sin\left(\frac{1}{2}(\mathrm{Am}_u - \mathrm{Am}_v)\right), \qquad\qquad f_{u,v} = \sec\left(\frac{1}{2}(\mathrm{Am}_u - \mathrm{Am}_v)\right). \qquad (A.4)$$

The $R$-matrix given in this appendix satisfies the Yang-Baxter equation. By using version 12.3 of Mathematica the check is straightforward, however, using version 12.0 particular attention should be payed to the choice of the sign of the branch-cut.

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
