# Peer review of "A range three elliptic deformation of the Hubbard model"

_SciPost Physics_

## Round 2 · Referee Report · Anonymous (Referee 1) · 2023-4-3

# Referee report on
## *A range three elliptic deformation of the Hubbard model*,
## by Marius de Leeuw, Chiara Paletta, Balázs Pozsgay

The authors introduce an integrable model with three-sites interaction, that can be viewed as a deformation of the Hubbard model. The paper is interesting and may be published, but there are some points that need to be clarified. In particular, the exposure on the integrability of the model for the three site interaction model (section 5 of the paper), looks rather unclear:

1. I think more details are needed to see that the eq. (5.13) ensures the integrability of the model. First, I think that eq. (5.12) should be $\check{\mathcal{L}}_{abj} = P_{bj}P_{aj}\mathcal{L}_{abj}$ (I suppose that they mixed with $\mathcal{L}_{abj} = P_{aj}P_{bj}\check{\mathcal{L}}_{abj}$). Second the equation $\check{R}_{12,34} = P_{13}P_{24}R_{12,34}$ should be added.

   It would be nice also to add that eq (5.17) comes from the relation $[\check{R}_{12}, \check{R}_{34}] = 0$.

2. I don't understand the discussion following eq. (5.16):

   - Starting with the Yang-Baxter eq. for $\check{R}$, one gets directly the relation (5.26), so I don't understand the whole discussion to get it.

   - It is easy to check that eq. (5.16) is obeyed using eqs (5.23) and (5.24), I don't see the need to introduce $\mathcal{G}$, specially if it is to conclude that it corresponds to $\mathcal{L}$.

   - After eq (5.22), what means direct calculation? Is it by using the assumption that $\mathcal{G}$ is $\mathcal{L}$? Then, again, all this is not needed since we get directly (5.26) from the Yang-Baxter eqs. If it is a general fact, I don't see how they get that. By the way, in relation (5.19), using the form (5.24), you find that $\mathcal{G}$ does not depend on $u_2$, but does depend on $u_3$.

   - A priori, eqs (5.23) and (5.24) do not define the same $R$ matrix, it is only because of eq. (5.26) that they match. It is true that there is a sentence about that in the paper, but the way it is written makes things look like (5.26) is deduced from this compatibility. I think this should be rephrased.

Apart from integrability, there are also minor points to be corrected

1. In section 2, when speaking of the symmetries of the Hubbard model, it is rather surprising to read that the periodic Hubbard model do not have $so(4)$ symmetry. I don't say what is written is wrong, it is just that the model they choose to be the Hubbard one is a bit unusual. To be fair, they indeed stress that, but I think it will be useful to remind it around eq. (2.34).

2. After eq (3.6), I would say that "The latter commutator vanish only in the case..." is closer to what they want to say.

3. Around eq (3.7) (special points), I think it would be clearer (and more precise) to say that $Q_2^\mu$ in eq (3.8) commutes with $\sum_j h_{j,j+1}^\mu$ and $\sum_j l_{j,j+1,j+2}^\mu$, separately ($\mu = \sigma, \tau$).

4. End of section 3: when the authors say that they tested all range 2 charges, does it include mixture of operators based on $\sigma$ and operators based on $\tau$? I suppose yes, but in view of the results, it would be clearer to state it explicitly.

5. Top of page 9, I don't understand why the odd number of down spin sector should not affect the thermodynamical properties of the model they study. Please detail the argument.

6. In section 4.2, when speaking of self-invariance, add somewhere 'up to a redefinition $j \to j - \frac{1}{2}$'. I am not sure that this redefinition makes the things clearer (one never knows of what model they speak of), but I leave the decision to the authors.

7. In eq (4.17), I think that $u = 8U \cos^4 \theta/2$.

8. Top of page 10, when speaking of actual Hubbard, do they mean any form of Hubbard model, or just their choice $H_1$ in eq (2.15)? Please specify it, and it would be more convincing if they tested the different forms of Hubbard.

9. After eq (4.21), you need to set $\alpha = 2Ue^{-2v}$ before the limit if you want the kinetic term to be kept.

10. In eq (5.6), I think a $log$ is missing (at least for the consistency of the exposure), and the sentences before and after this eq. should be merged.

11. Section 5.2, when saying that the higher charges are local, it should be specified that the locality is not of nearest neighbors type, but is of longer range, and depends on the charge. It is local only in the sense that they remain finite, even in the limit $L \to \infty$.

12. In eq (5.11), is it not $Tr_{ab}$ that you should have?

13. In appendix A, it would be nice to have also an expression of $R$ (or $\check{R}$) in term of $\sigma$ matrices. That would help the interested reader to compute $\mathcal{L}_{j,j+1,j+2}$.

---

## Round 2 · Referee Report · Niklas Beisert (Referee 2) · 2023-4-13

Report

The submitted manuscript introduces two integrable modifications of the 1-dimensional Hubbard model. One represents a deformation with interactions ranging over three sites, the other consists of standard nearest-neighbour interactions which, however, do not preserve spin. These two models are related by a bond-site transformation, and their underlying R-matrix is derived. It depends on both spectral parameters (individually) through elliptic functions, and it represents a novel solution of the Yang-Baxter equation. These results leading to a new and very non-trivial quantum R-matrix exciting, and they call for further exploration and classification. For these reasons I recommend publication in SciPost.

Requested changes

In the following I list some remarks that the authors should address towards publication:

1) Brackets in equation (2.4) seem to be missing

2) Above equation (2.15), Identity is spelled with a capital I

3) The paragraph above equation (3.4) mentions a new formalism for integrability of medium-range interaction developed in reference [32] and to be reviewed further in paragraph 5.2. It may help the reader to already here sketch some key element(s) of this formulation, i.e. what is this about?

4) equation (5.6) seems to be missing a log as in (5.5) which is also mentioned just below the equation.

5) I don't quite follow the claims around (5.5): It seems that regularity of the Lax operator would ensure a local Hamiltonian, but I do not see how regularity of the R-matrix would help.

6) Equation (5.8) defines the relationship between L and R with an unusual dependency on the two parameters. Does this work generically or specifically in this model and/or with the particular transformation of the parameters

7) Together items 6) and 7) might work, but taking the relationship of parameters into account shouldn't one consider the point u=mu/alpha(mu) for L(u,mu) such that R is evaluated at R(mu, mu)?

8) It is worthwhile to point out that the product/quotient of the two quantities (5.10) is an elliptic functions. Hence, only one of the two escapes the elliptic property. Similarly, it would suffice to use just one of the functions f and g in appendix A.

9) Is the expression in equation (5.12) really correct? For a standard flow of the tensor sites, one that is naturally compatible with equation (5.13), the two permutation operators Paj and Pbj should be exchanged. Then effectively Lchech_abj would map site j to site a while ab are shifted to bj. Equation (5.13) would then be a rather natural YBE in the R/Lchech notation with two pairs of sites and a single site. As it stands, Lchech_abj seems to shift a to j and bj to ab, which is the opposite cyclic permutation for what is needed to make (5.13) work out naturally. Finally, defining the Hamiltonian to be a logarithmic derivative, would (naturally) yield local terms only if the correct permutation is assumed.

10) Also (5.17) appears rather odd. Is this specific to the model, or does it follow generically from the bond-site transformation? If so, how?

11) The following relations all appear a bit unnatural, could be related to items 9) and/or 10). However, in the end the authors obtain a rather natural YBE in equation (5.27) for three pairs of sites. Now I'm not really questioning the validity of (5.12) through (5.27) which I have not checked in terms of expressions, but I'm wondering whether it is a coincidence that they hold, and whether a simpler set of relations (based on the permutation in item 9) would (also) hold. A less technical and more descriptive exposition of the would also avoid such potential misunderstandings.

12) Is equation (5.27) the principal outcome of its section? It would seem more straight-forward to derive it as the compatibility condition for equation (5.13).

13) Below equation (5.22): grammar at "depends" and "in" appears wrong.

14) Beginning of section 6): How to combine these two limits? Are these two different limits, two separate limits to be taken in some (any?) order or one simultaneous limit? If so, how to write u and U as limit of a common parameter?

15) Appendix A promises the computer algebra expression of the R-matrix upon request to the authors. For a modern digital publication, it would seem more natural to supply the expression as ancillary digital material to the publication as it represents relevant scientific data. This could be done easily with a minimal mathematica notebook containing the expressions of appendix A in machine readable format.

---

## Round 2 · Referee Report · Takuya Matsumoto (Referee 3) · 2023-4-29

Title: `A range three elliptic deformation of the Hubbard model`
Arxiv Link: https://arxiv.org/abs/2301.01612v2
Authors: Marius de Leeuw, Chiara Paletta, Balázs Pozsgay

# Referee Report

This manuscript proposes an integrable deformation of the 1D Hubbard models with the consecutive three sites interactions and violating the particle number conservation. The Hamiltonian $H_3$ includes the two coupling constants $u$ and $\kappa$, where $u$ is the coupling constant of the Hubbard model and $\kappa$ is that of the three-range interaction. Sec.2 reviews the 1D Hubbard models so that the formulation fits the author's purpose. In Sec.3, the Hamiltonian $H_3$, including the three range interactions, are introduced. After the non-triviality and the characteristic feature of $H_3$ are discussed, the integrability is proved. In Sec.4, using the bond-site transformation, the authors convert the three-range interaction to the nearest-neighborhood interaction and define a new two-site model. In Sec.5, the two-site and three-site R-matrices are derived. The authors show their integrability in the sense that they satisfy the YBE. The explicit form of the two-site R-matrix is presented in App. A. They observe that the matrix entries of the two-site R-matrix are elliptic functions of the spectral parameters. Sec.6 argues the particular limits with respect to the Hubbard coupling $u$ and $U$.

This article introduces the interesting three-range Hamiltonian $H_3$. Then, using a bond-site transformation effectively, the authors find the new elliptic solution of the YBE. These results are new and interesting. Thus, I recommend this manuscript for publication in SciPost. However, before going to the publication, I would appreciate the author providing further clarifications and expositions on the following points and improving your manuscript appropriately.

1. (2.1) seems to lack the commutation relations among the creation operators; $\{c^\dagger, c^\dagger\}$.

2. In this paper, I think that all notations $SU(2)$ and $U(1)$ stand for the Lie algebra rather than the Lie group. The Lie algebras are usually expressed by small letters such as $\mathfrak{su}(2)$ and $\mathfrak{u}(1)$. Hence, I am afraid that the capital notations are a bit confusing to readers with pure mathematical backgrounds.

3. From (2.21) to (2.27), the authors show that two sets of the generators $A$'s and $B$'s satisfy the $\mathfrak{su}(2)$ algebras, respectively. For completeness, it is worth adding the commutation relations among them, i.e., $[A, B] = 0$, which concludes the whole algebra is $\mathfrak{su}(2)\oplus\mathfrak{su}(2)$.

4. Concerning the three-range Hamiltonian $H_3$ in (3.2), I don't see how the authors arrived at this solution. For instance, why doesn't it have $(j, j + 2)$ interaction? Why does the

three-range interaction include only $\sigma^x$, $\sigma^z$ but $\sigma^y$? Some heuristic arguments would help readers.

5. The authors claim that the model given by $H_3$ is integrable in the sense that it has infinitely many commuting conserved charges. What do the authors mean by a transfer matrix is $t(u)$ in (5.11)? If so, I expect that the integrability here means $[t(u), t(v)] = 0$. Then, by canonical arguments, the commuting charges are obtained by expanding $t(u)$ with respect to $u$. Since the proof of integrability of $H_3$ is essential, I suggest authors to reproduce the formulation of [32] along this situation.

6. Relating to the above comment, it would be nice if authors could present the explicit expressions of the first few non-trivial conserved charges appearing in the expansion of $t(u)$.

7. The authors state that the two-site Hamiltonian $H_2$ in (4.8) does not include the actual Hubbard model for any choice of $\theta$. Is there more conceptual exposition for this point? Since $H_3(\kappa)$ reduces to the Hubbard Hamiltonian $H_3(0)$ at $\kappa \to 0$, I think that it is natural to expect $H_2(\kappa)$ to become $H_2(0) = H_3(0)$ at the same limit. If not, mathematically, a bond-site transformation seems not to preserve the continuity of $\kappa$ at $\kappa = 0$, i.e., $\lim_{\kappa \to 0} H_2(\kappa) \neq H_2(0)$. Why does this happen? Does this phenomenon relate to parameterization (4.7)?

8. In the second line of Sec.5, it is written as "Quantum Inverse Scattering Approach." But, I feel that "Quantum Inverse Scattering Method (QISM)" is more common. It is not a request but my suggestion for consideration.

9. In (5.6), I think either $t^{-1}$ or log is missing.

10. In (5.12), $\check{L}$ should also have three subscripts. In [32], $\check{L}$ is defined as $L = PP\check{L}$. Do authors use a different convention here?

11. (A.1) presents the explicit form of the R-matrix. Mathematically, the matrix expression only makes sense by specifying the ordering of the basis and the action of the liner operator on each base. These two points should be clarified.

12. Relating to 7., doesn't the R-matrix (A.1) reduce to the R-matrix of trigonometric type in [24] in any limit of $\kappa$?

13. Throughout the manuscript, the terms two/three-site and 2/3-site are used. It would be better to integrate these into one of them.

---

## Editorial Decision

resubmitted